# Preparation and Evaluation of Inhalable Amifostine Microparticles Using Wet Ball Milling

**DOI:** 10.3390/pharmaceutics15061696

**Published:** 2023-06-09

**Authors:** Jae-Cheol Choi, Ji-Hyun Kang, Dong-Wook Kim, Chun-Woong Park

**Affiliations:** 1College of Pharmacy, Chungbuk National University, Cheongju 28160, Republic of Korea; chlwocjf77@gmail.com; 2School of Pharmacy, Jeonbuk National University, Jeonju 54896, Republic of Korea; jhkanga@hanmail.net; 3College of Pharmacy, Wonkwang University, Iksan 54538, Republic of Korea; pharmengin@gmail.com

**Keywords:** amifostine trihydrate, polar non-polar solvent, ball mill, dry powder inhaler, pulmonary drug delivery

## Abstract

The conventional dosage form of Ethyol^®^ (amifostine), a sterile lyophilized powder, involves reconstituting it with 9.7 mL of sterile 0.9% sodium chloride in accordance with the United States Pharmacopeia specifications for intravenous infusion. The purpose of this study was to develop inhalable microparticles of amifostine (AMF) and compare the physicochemical properties and inhalation efficiency of AMF microparticles prepared by different methods (jet milling and wet ball milling) and different solvents (methanol, ethanol, chloroform, and toluene). Inhalable microparticles of AMF dry powder were prepared using a wet ball-milling process with polar and non-polar solvents to improve their efficacy when delivered through the pulmonary route. The wet ball-milling process was performed as follows: AMF (10 g), zirconia balls (50 g), and solvent (20 mL) were mixed and placed in a cylindrical stainless-steel jar. Wet ball milling was performed at 400 rpm for 15 min. The physicochemical properties and aerodynamic characteristics of the prepared samples were evaluated. The physicochemical properties of wet-ball-milled microparticles (WBM-M and WBM-E) using polar solvents were confirmed. Aerodynamic characterization was not used to measure the % fine particle fraction (% FPF) value in the raw AMF. The % FPF value of JM was 26.9 ± 5.8%. The % FPF values of the wet-ball-milled microparticles WBM-M and WBM-E prepared using polar solvents were 34.5 ± 0.2% and 27.9 ± 0.7%, respectively; while the % FPF values of the wet-ball-milled microparticles WBM-C and WBM-T prepared using non-polar solvents were 45.5 ± 0.6% and 44.7 ± 0.3%, respectively. Using a non-polar solvent in the wet ball-milling process resulted in a more homogeneous and stable crystal form of the fine AMF powder than using a polar solvent.

## 1. Introduction

Amifostine (AMF), also known as Ethyol^®^, is an organic thiophosphate extensively studied as a cytoprotective agent [1]. It is used as a cytoprotective agent against the damaging effects of ionizing radiation and chemotherapy and has been approved by the U.S. Food and Drug Administration [1,2]. The current use of AMF in clinical practice is limited to reducing the cumulative renal toxicity associated with repeated cisplatin administration in patients with advanced ovarian or non-small-cell lung cancer [3,4]. AMF (Ethyol^®^) administered to cancer patients is rapidly cleared from plasma by a biphasic decay with an alpha half-life (T1/2α) of 0.88 min and a T1/2β of 8.8 min [1]. Therefore, AMF is administered to patients as a 15 min intravenous infusion before cisplatin and carboplatin administration to protect against dose-limiting toxicities [1,5]. However, AMF continues to be evaluated as a cytoprotector in other clinical settings involving radiotherapy and chemotherapy [2,6,7,8]. Targeted pulmonary delivery of topical therapeutically active pharmaceutical ingredients via inhalation is an effective and non-invasive method [9]. Among the different types of targeted pulmonary delivery formulations and devices, dry-powder inhalers (DPIs) offer several advantages, including high stability in the dry state, ease of handling, and portability [10,11]. DPIs are propellant-free, portable, and physicochemically stable compared with pressurized metered-dose inhalers, soft-mist inhalers, and nebulizers [12,13]. Pulmonary delivery of a small dose of AMF can be achieved by minimizing systemic side effects due to its low absorption into the body [13]. The efficiency of a DPI is determined by the aerosol performance of the particles and the physicochemical properties of the formulation, including the density, shape, particle size distribution, surface morphology, interparticle forces, and solid state [3,14]. The aerosolization efficiency of a DPI is expressed as the aerodynamic diameter of the particle, with 1–5 µm being considered respirable for deep pulmonary delivery [15]. Particles of 5 μm or less are deposited in bronchial tubes, while particles of 1 μm reach the alveoli [9,16]. Therefore, particle micronization is important for lung drug delivery to achieve the desired target deposition of therapeutic agents. Various types of mills can be used to reduce the particle size. Inhalable microparticles are mostly engineered using top-down and bottom-up processes, depending on the starting material. Milling is used to process powders in the pharmaceutical industry, particularly for inhalation drugs and carriers [14,17,18]. However, milling often induces disorder/defects on the surface of particles, resulting in changes in crystallinity [19]. These defects and changes in crystallinity may adversely affect the aerodynamic performance and stability of the drug [20,21,22]. The crystalline form of the drug varies with hydration, resulting in differences in the dissolution rate and solubility, and affecting bioavailability [23]. Purification of a stable crystalline form is a steady process with good reproducibility [24]. Jet milling is a process of reducing the particle size by particle–wall collisions using high-velocity jets [25]. In the jet mill, the material passes through a vibration feeder into the grinding chamber. The particles are accelerated by air jets passing through the pusher nozzles, and the high pressure and velocity of the air flowing through the two grinding nozzles cause the material to experience extreme turbulence. Wet ball milling (WBM) is a process of size reduction using bead–particle collisions in a rotating vessel, enabling the milling of heat-sensitive materials and contamination-free refinement [26]. The mechanical parameters of ball milling, ball-mill solvent, and rotation time strongly influence the physicochemical properties of the particles [27]. The influence of solvent polarity is also important in ball milling. Organic reactions and recrystallization are significantly affected by the polarity of the compound molecule and the solvent [28]. Methanol, ethanol, chloroform, and toluene were selected as solvents to prepare inhalable AMF microparticles. This study aimed to develop inhalable AMF microparticles with excellent inhalation efficiency and stability. Fine particles were prepared using WBM under different solvent conditions (polar and non-polar), and their physicochemical properties, drug release, and drug inhalation delivery efficiency were evaluated [29,30,31,32].

## 2. Materials and Methods

### 2.1. Materials

AMF (C5H15N2O3PS, molecular weight (MW) 214.23 g/mol) was purchased from Tianjin Zhongrui Pharmaceuticals Co., Ltd. (Tianjin, China). HPLC-grade methanol, ethanol, chloroform, toluene, and acetonitrile (Honeywell, Burdick & Jackson, Muskegon, MI, USA) were of HPLC grade. All experiments were performed using Milli-Q distilled water. All the other chemicals were of reagent grade.

### 2.2. Preparation of Inhalable AMF Microparticle Using Jet Milling

AMF microparticles were prepared using jet milling (JM) (A-O Jet mill, JS TECH, Cheonan City, Republic of Korea) using the following parameters: G nozzle (MPa), 0.4; P nozzle (MPa), 0.6.

### 2.3. Preparation of Inhalable AMF Microparticle Using Wet Ball Milling

To prepare the AMF microparticles, the collision medium consisted of spherical zirconia (ZrO_2_) balls of a single size of 3.00 mm (PM 100, Retsch Co., Ltd., Düsseldorf, Germany). Grinding was performed as follows: AMF (10 g) and zirconia balls (50 g) were mixed and placed in a cylindrical stainless-steel jar lined with ZrO_2_; a 56 cm^3^ jar was filled with an 80 vol% of particles and balls. To investigate the influence of the solvent, 20 mL each of methanol, ethanol, chloroform, and toluene were added and prepared using the same procedure. The ball milling was performed at 400 rpm for 15 min. The direction of jar rotation was set counter to that of the disk revolution. Moreover, it was reported in the literature that a significant grinding rate was obtained when the mill jar was rotated near the critical speed ratio counter-directionally to the disk revolution, which was also rotated near the critical speed ratio [33]. After grinding the ball mill, the solvent was removed by filtration under reduced pressure with a 0.45 μm membrane filter and drying. The samples were judged to be sufficiently dried after drying overnight at 60 °C, which is near the vaporization temperature of the solvents. As the solvents used are highly volatile and drying was carried out for an extended period, very few residual solvents are expected [34,35] (Figure 1).

### 2.4. Physicochemical Properties of Inhalable AFM Microparticles

#### 2.4.1. Scanning Electron Microscopy (SEM)

Visual imaging of raw AMF and milled formulations (AMF, jet-milled (JM) microparticles, WBM-M, WBM-E, WBM-C, and WBM-T) was performed using a scanning electron microscope (SEM; ZEISS-GEMINI LEO 1530, Zeiss, Jena, Germany). The samples were placed onto carbon tape and coated with a 200 Å thickness of platinum using a Hummer VI sputtering device and then imaged at a voltage of 3 kV and magnifications of 500× and 10,000×.

#### 2.4.2. Particle Size Distribution (PSD)

Particle size distribution was determined by laser diffraction particle sizing (Mastersizer 2000, Malvern Instruments, Worcestershire, UK) using the wet dispersion method after dispersing the samples in toluene.

#### 2.4.3. Powder X-ray Diffraction (PXRD)

Powder X-ray diffraction (PXRD) patterns were obtained using an X’Pert PRO MRD^®^ diffractometer (PANAlytical Ltd., Almelo, The Netherlands) with Cu K radiation at 50 mA and 40 kV. The AMF and AMF microparticles (JM, WBM-M, WBM-E, WBM-C, and WBM-T) were placed on a plate at room temperature. The 2 h scans were performed between 5° and 60° at a scan interval of 0.1°.

#### 2.4.4. Thermo Gravimetric Analysis (TGA)

TGA was performed using an SDT 2960 instrument (TA Instruments Ltd., New Castle, DE, USA). AMF, JM, WBM-M, WBM-E, WBM-C, and WBM-T samples weighing approximately 1 mg each were placed in aluminum pans and hermetically sealed. The samples were heated from 25 to 300 °C at a heating rate of 10 °C/min. All experiments were conducted in triplicate.

#### 2.4.5. Differential Scanning Calorimetry (DSC)

Differential scanning calorimetry (DSC) was performed using a Q2000 instrument (TA Instruments Ltd., New Castle, DE, USA). AMF, JM, WBM-M, WBM-E, WBM-C, and WBM-T samples weighing approximately 1 mg each were placed in aluminum pans and hermetically sealed. The samples were heated from 0 to 300 °C at a heating rate of 10 °C/min. All experiments were conducted in triplicate.

#### 2.4.6. Fourier Transform Infrared (FT-IR) Spectroscopy

FT-IR was performed on AMF, JM, WBM-M, WBM-E, WBM-C, and WBM-T from 500 to 4000 cm^−1^ using an IFS 66/S^®^ spectrometer (BRUKER OPTIK GMBH Ltd., Billerica, MA, USA).

#### 2.4.7. Karl Fischer Titration

Measurements were performed on cells without a diaphragm, using a C30 compact Karl Fischer coulometer (Mettler Toledo, Billerica, MA, USA). To determine the residual water content of AMF, JM, WBM-M, WBM-E, WBM-C, and WBM-T, a titrator (Mettler Toledo, C30SX, Greifensee, Switzerland) was used.

### 2.5. In-Vitro Aerodynamic Performance

#### Next-Generation Impactor (NGI)

In accordance with the USP Chapter <601> specification for aerosols, the aerosol performance of the formulation was determined using a seven-step non-polymer next-generation impactor (Model 170, COPLEY Scientific Limited, Nottingham, UK) and Handihaler^®^ (Boehringer Ingelheim, Ingelheim am Rhein, Germany) DPI devices. The airflow rate was measured using a flowmeter (DFM 2000, COPLEY Scientific Limited, Nottingham, UK) operated at a control flow rate of 60 L/min using a vacuum pump. The collection plates of the NGI stage were precoated with silicone oil to prevent bouncing and re-entrainment of the particles. Hard hydroxypropyl methylcellulose capsules (size 4) were loaded with 10 mg of the drug. A capsule was inserted into the RS-01, and the device was inserted into the mouthpiece of the induction port. Air was inhaled at a controlled flow rate of 60 L/min for 4 s. For an NGI flow rate of 60 L/min, the aerodynamic cutoff diameters of each stage were determined as 8.06 µm, 4.46 µm, 2.82 µm, 1.66 µm, 0.94 µm, 0.55 µm, and 0.34 µm for stages 1–7. The quantity of particles deposited on each collector plate of the stage was measured using a validated HPLC, as described below. The AMF in the extended microparticles were analyzed using an HPLC system (Thermo Fisher Scientific Co., Ltd., Waltham, MA, USA) consisting of an isocratic pump (Ultimate 3000), an autosampler (Ultimate 3000), a column oven (Ultimate 3000), a UV detector (Ultimate 3000), and HPLC System software with a detection wavelength of 220 nm. The analytical column used was an Aegispak C8-L 5 µm, 4.6 mm × 250 mm. The mobile phase consisted of a solution of 0.94 g/L of sodium 1-hexanesulfonate adjusted with phosphoric acid to a pH of 3.0: methanol (72:28 (*v*/*v*)) delivered at a rate of 1 mL/min. The column temperature was maintained at 5 °C and the injection volume of each sample was 10 µL. The fine particle fraction (FPF) indicates the ability of particles to reach the respirable region. It consists of particles with an aerodynamic size of about 5.0 μm or less and is expressed as the mass percentage collected at stages 1–7. The emitted dose (ED) is the percentage difference between the initial mass and the remaining mass of the particles in the capsules after aerosolization of the initial mass. The equations for FPF and ED are as follows [36,37,38]:Fine particle fraction (FPF)% = mass of particles in stages 1 through 6/emitted dose (ED) × 100
Emitted dose (ED)% = (initial mass in capsule − final mass remaining in capsule) initial mass in capsule × 100

The mass median aerodynamic diameter (MMAD) and the geometric standard deviation (GSD) were calculated by the USP method. All experiments were performed in triplicate (*n* = 3).

## 3. Results

### 3.1. Physicochemical Properties

#### 3.1.1. Particle Characterization and Morphology

Scanning electron micrographs (SEM) of raw and milled AMF are shown in Figure 2. The raw AMF were close to the plate and regular in shape, and the JM microparticles and non-polar-solvent wet-ball-milled microparticles (WBM-C, WBM-T) were smaller in size in the raw AMF; however, the polar-solvent wet-ball-milled microparticles (WBM-M, WBM-E) had irregular shapes that were aggregated. In terms of size, the polar-solvent wet-ball-milled microparticles (WBM-M and WBM-E) showed polydispersity, whereas the non-polar-solvent wet-ball-milled microparticles (WBM-C and WBM-T) were monodisperse. The polar-solvent wet-ball-milled microparticles (WBM-M and WBM-E) exhibited rough surfaces. The JM and non-polar-solvent wet-ball-milled microparticles (WBM-C and WBM-T) had similar surface shapes and were characterized by a relatively smoother surface than that of the polar-solvent wet-ball-milled microparticles (WBM-M and WBM-E).

#### 3.1.2. Particle Size Distribution (PSD)

The PSD of the raw AMF, JM microparticles, polar-solvent wet-ball-milled microparticles (WBM-M and WBM-E), and non-polar-solvent wet-ball-milled microparticles (WBM-C and WBM-T) are as follows: The raw AMF microparticles were of 28.0 µm (Dv 10), 89.2 µm (Dv 50), and 181 µm (Dv 90) in size with a span value of 1.712. The JM microparticles were 2.91 µm (Dv 10), 7.2 µm (Dv 50), and 15.4 µm (Dv 90) in size with a span value of 1.740. The polar-solvent wet-ball-milled microparticles (WBM-M, WBM-E) were 1.06 µm, 1.63 µm (Dv 10), 6.0 µm, 5.76 µm (Dv 50), 23.6 µm, and 16.3 µm (Dv 90) in size with span values of 3.753 and 2.541. The non-polar-solvent wet-ball-milled microparticles (WBM-C, WBM-T) were 2.01 µm, 2.33 µm (Dv 10), 5.95 µm, 5.36 µm (Dv 50), 13.5 µm, and 11.3 µm (Dv 90) in size with span values of 1.924 and 1.681. Jet-milled and ball-milled microparticle formulations were manufactured in an inhalable size with a Dv50 of 5–7 µm. The AMF, JM microparticles, and non-polar-solvent wet-ball-milled microparticles (WBM-C and WBM-T) exhibited similar span values with a monodisperse distribution. In contrast, the polar-solvent wet-ball-milled microparticles (WBM-M and WBM-E) exhibited high span values and polydispersity.

#### 3.1.3. Powder X-ray Diffraction (PXRD)

The diffractograms of the JM microparticles and non-polar-solvent wet-ball-milled microparticles (WBM-C and WBM-T) were almost identical, with sharp diffraction peaks at 8.24°, 13.32°, 16.46°, 21.12°, 23.62°, 24.52°, and 26.73°, indicating their high crystallinity. The intensities of the principal diffraction peaks of the JM microparticles and non-polar-solvent wet-ball-milled microparticles (WBM-C and WBM-T) were significantly lower than those of the AMF, but most peak positions were similar. During the milling process, the particle size and intensity of the peaks decreased. In contrast, the polar-solvent wet-ball-milled microparticles (WBM-M and WBM-E) did not show the same main peaks in the AMF. The main diffraction peaks were at 8.70°, 12.66°, 15.06°, 17.42°, 20.24°, 22.06°, 22.82°, and 29.10°. This result suggests that they have completely different crystalline forms. The milling process affected the crystalline form. Changes in crystal polymorphism were observed in the formulations using polar solvents (WBM-M and WBM-E), which also affected the inhalation efficiency of the particles (Figure 3).

#### 3.1.4. Thermal Gravimetric Analysis (TGA)

The TGA diagrams are shown in Figure 4. In the presence of the trihydrate of AMF, the weight decreased by approximately 21% at 100 °C. Comparing the molar mass of AMF (214.22 g·mol^−1^) and that of the trihydrate (54.05 g·mol^−1^) gives a water mass of 20.1%. The mass reduction patterns of JM, WBM-C, and WBM-T were similar to those of raw AMF. However, WBM-M and WBM-E had a mass decrease of only approximately 8% at 100 °C. This result indicates that dehydration affected the preparation of the polar-solvent wet-ball-milled microparticles (WBM-M, WBM-E). 

#### 3.1.5. Differential Scanning Calorimetry (DSC)

DSC thermograms of Raw AMF, JM, WBM-M, WBM-E, WBM-C, and WBM-T were observed in the range of 50–230 °C. The melting point of AMF was found to be 149 °C, and the endothermic peak of trihydrate was seen at 91.4 °C. The melting point of the AMF in JM was found to be 144 °C, and the endothermic peak of the trihydrate was seen at 93.4 °C. The AMF melting points in WBM-M and WBM-E were observed at 144 °C and 140 °C, respectively. However, the endothermic peak of trihydrate was not observed, suggesting that hydrate removal occurred during the ball-milling process. The melting points of AMF in WBM-C and WBM-T were measured to be 147 °C and 142 °C, respectively (Figure 5).

#### 3.1.6. Fourier Transform Infrared (FT-IR)

In the AMF, the presence of a hydrogen-bond peak of the hydrate in the FT-IR spectrum represents the interaction between atoms or molecules that are electrically bonded to hydrogen atoms. When moisture is removed during dehydration, the hydrogen bonds disappear, resulting in a change in peak shape or intensity, or a peak shift. Hydrogen-bond peaks appear mainly between the wavenumbers of 1500–4000 cm^−1^ in the infrared spectrum. Within this range, hydrogen-bond peaks can appear in various forms and generally exhibit characteristics within the following wavenumber range: The ester group (C=O) easily forms hydrogen bonds with oxygen atoms. The peaks corresponding to these hydrogen bonds were observed at 1738.4 cm^−1^. In addition, a peak associated with OH stretching was observed at 3396.4 cm^−1^. JM, WBM-C, and WBM-T exhibited peaks similar in position, intensity, and shape to those of AMF. In contrast, there were changes in the peaks of WBM-M and WBM-E. The shape of the 1738.4 cm^−1^ peak associated with the ester group has changed. In addition, the strength and shape of the 3396.4 cm^−1^ peak associated with OH stretching were clearly changed. This suggested changes in the molecular structure of the particles due to dehydration [39] (Figure 6).

#### 3.1.7. Karl Fischer

The free surface moisture or moisture contained in the crystal was measured using a Karl Fischer instrument. In the AMF, JM, and non-polar-solvent wet-ball-milled microparticles (WBM-C and WBM-T), similar moisture contents were observed (21.8, 22.8, 22.6, and 22.7%). In contrast, the polar-solvent wet-ball-milled microparticles (WBM-M, WBM-E) had a low moisture content (12.1, 12.4%). The water content of the AMF trihydrate was 21.8%.

### 3.2. Aerodynamic Characterization

#### Next Generation Impactor (NGI)

Aerosol performance was characterized using the NGI and RS-01^®^ devices. The % ED values of all formulations ranged from 89.1 to 93.5%, indicating a low dose loss in the capsule during inhalation. The % FPF value was not measured for the raw AMF. The % FPF values at initial sampling were 26.9 ± 5.8%, 34.5 ± 0.2%, 27.9 ± 0.7%, 45.5 ± 0.6%, and 44.7 ± 0.3% for JM, WBM-M, WBM-E, WBM-C, and WBM-T, respectively. The MMAD values for the sampling of JM, WBM-M, WBM-E, WBM-C, and WBM-T were 4.0 ± 0.1 µm, 3.0 ± 0.1 µm, 2.8 ± 0.1 µm, 3.3 ± 0.1 µm, and 3.1 ± 0.1 µm, respectively. The GSD values for the initial sampling of JM, WBM-M, WBM-E, WBM-C, and WBM-T were 1.9 ± 0.1 µm, 2.8 ± 0.1 µm, 2.9 ± 0.1 µm, 2.8 ± 0.1 µm, and 2.7 ± 0.1 µm, respectively. The non-polar-solvent wet-ball-milled microparticles (WBM-E and WBM-T) had a low deposit amount in the induction port; thus, the inhalation efficiency was measured to be high. In addition, many deposits were observed in the cup stage. This suggests that the drug was delivered deep into the lungs. The evaluation of the physical properties showed that the polar-solvent wet-ball-milled formulations (WBM-M and WBM-E) were in an unstable form (dehydrated), suggesting that the inhalation efficiency was degraded due to moisture absorption and aggregation between particles. Conversely, the non-polar-solvent wet-ball-milled microparticles (WBM-C and WBM-T) had similar physical properties to those of the AMF, and a high inhalation efficiency was confirmed (Figure 7).

## 4. Discussion

In this study, we aimed to prepare and evaluate inhalable microparticles as an alternative to injectable formulations containing AMF. As shown in Figure 1 and Table 1, wet ball-milling conditions were determined. To determine the inhalable particle size of AMF, jet milling using air, wet ball milling in polar solvents (methanol and ethanol), and wet ball milling in non-polar solvents (chloroform and toluene) were performed. The wet ball-milling process was performed as follows: AMF (10 g), zirconia balls (50 g), and solvent (20 mL) were mixed and placed in a cylindrical stainless-steel jar. Wet ball milling was performed at 400 rpm for 15 min.

First, to examine the characteristics of the prepared samples, SEM and PSD measurements were performed. The morphologies of the ground particles were visually observed using SEM. However, particle aggregation was confirmed in the WBM-M and WBM-E formulations. The PSD of the prepared samples were as follows: The raw AMF microparticles were 28.0 µm (Dv 10), 89.2 µm (Dv 50), and 181 µm (Dv 90) in size with a span value of 1.712. The JM microparticles were 2.91 µm (Dv 10), 7.2 µm (Dv 50), and 15.4 µm (Dv 90) in size with a span value of 1.740. The polar-solvent wet-ball-milled microparticles (WBM-M, WBM-E) were 1.06 µm, 1.63 µm (Dv 10), 6.0 µm, 5.76 µm (Dv 50), 23.6 µm, and 16.3 µm (Dv 90) in size with span values of 3.753 and 2.541. The non-polar-solvent wet-ball-milled microparticles (WBM-C, WBM-T) were 2.01 µm, 2.33 µm (Dv 10), 5.95 µm, 5.36 µm (Dv 50), 13.5 µm, and 11.3 µm (Dv 90) in size with span values of 1.924 and 1.681. The results of the PSD show that particles of sufficient inhalable size were prepared after milling. However, the WBM-M and WBM-E formulations had high Dv 90 and wide span values. This suggests that aggregation occurred between the particles. The JM microparticles and non-polar-solvent wet-ball-milled microparticles (WBM-C and WBM-T) showed partially similar crystallinity to that of raw AMF. The main peaks had (2θ) angles of 8.24°, 13.32°, 16.46°, 21.12°, 23.62°, 24.52°, and 26.73°. The intensity of the main peaks decreased, but their positions did not shift. This suggests that they have similar crystalline forms. However, the main peaks of the polar-solvent wet-ball-milled microparticles (WBM-M, WBM-E), which were at (2θ) angles of 8.70°, 12.66°, 15.06°, 17.42°, 20.24°, 22.06°, 22.06°, 22.82°, and 29.10°, show completely different shapes and positions. This result suggests that they have completely different crystalline forms.

In the TGA (Figure 4) results, the reduced weight (%) at 100 °C indicates the presence of hydrates. In the presence of the trihydrate of AMF, the weight decreased by approximately 21% at 100 °C. However, in the dehydrated WBM-M and WBM-E formulations, only 8% of the weight (%) was lost. The AMF melting points in WBM-M and WBM-E were observed to be 144 °C and 140 °C, respectively. However, the endothermic peak of trihydrate was not observed, suggesting that hydrate removal occurred during the ball-milling process. The FT-IR spectrum indicated a shift in the 3396.4 cm^−1^ peak related to -OH stretching, owing to the dehydration of hydrates. The shape of the 1738.4 cm^−1^ peak associated with the ester group has changed (Figure 6). Therefore, the polar-solvent wet-ball-milled microparticles (WBM-M and WBM-E) showed completely different crystallinities and trihydrate content to raw AMF. Through physicochemical evaluations, the dehydration of the hydrates of polar-solvent wet-ball-milled microparticles (WBM-M, WBM-E) was demonstrated. The trihydrate in the AMF exhibited stable crystallinity. The dehydration of hydrates results in an unstable crystalline form and negatively affects inhalation stability.

The results of the aerodynamic performance tests showed that the non-polar-solvent wet-ball-milled microparticles (WBM-C, WBM-T) had better inhalation efficiency than the polar-solvent wet-ball-milled microparticles (WBM-M, WBM-E). The % FPF value was not measured for the raw AMF. The % FPF values were 26.9 ± 5.8%, 34.5 ± 0.2%, 27.9 ± 0.7%, 45.5 ± 0.6%, and 44.7 ± 0.3% for JM, WBM-M, WBM-E, WBM-C, and WBM-T, respectively. Wet ball milling was more efficient for inhalation than jet milling. Of all the formulations, the formulations using non-polar solvents (WBM-C and WBM-T) were the best.

The AMF trihydrate helps maintain crystallinity in a stable form. Wet ball milling with a polar solvent resulted in reduced hydrate content. The reduction in hydrate content negatively affects crystallinity and inhalation efficiency. In ball milling, the crystal form is affected by changes in the hydrate content depending on the polarity of the solvent. In addition, as the unstable form (dehydrated) is transformed into a stable form (trihydrate), agglomeration due to the hygroscopicity of free water decreases the inhalation efficiency. Therefore, non-polar-solvent wet ball milling with stable crystallinity and inhalation efficiency is suitable for manufacturing inhalable particle sizes.

## 5. Conclusions

To prepare AMF particles, wet ball milling has been demonstrated to have a better inhalation efficiency than jet milling. In the wet-milling process, the crystalline form changes depending on the polarity of the solvent, which negatively affects the physicochemical characteristics, stability, and inhalation efficiency. Changes in the crystalline form and molecular structure of the hydrate due to dehydration were confirmed experimentally. The formulations (WBM-M and WBM-E) prepared using polar solvents were consistent with the SEM results showing significant aggregation between particles, and the PSD results showed a large average particle size and wide distribution. Additionally, the main peaks (2θ) in the PXRD patterns corresponded to a completely different crystalline form. The moisture content, weight loss, disappearance of the endothermic peak, and movement of the –OH stretching peak of the hydrate were measured. In addition, the inhalation efficiencies of the wet-ball-milled formulations were higher than that of the jet-milled formulation. Of all the formulations, the formulations using non-polar solvents (WBM-C and WBM-T) were the best. In wet ball milling, formulations using non-polar solvents (WBM-C and WBM-T) had higher inhalation efficiency than those using polar solvents (WBM-M and WBM-E) because of their small particle size and narrow distribution (low span value). During wet ball milling, the solvent influenced the changes in the hydrate content. Therefore, when preparing inhalable AMF microparticles using wet ball milling, using non-polar solvents minimized the change in hydrate content and enabled the manufacturing of high-efficiency inhalation formulations.

## Figures and Tables

**Figure 1 pharmaceutics-15-01696-f001:**
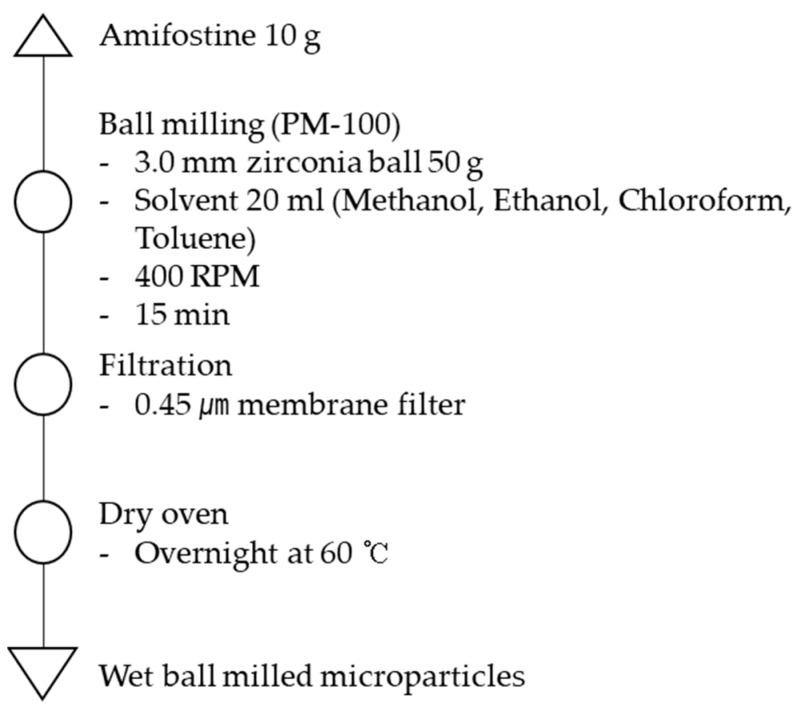
Schematic diagram of the wet ball-milling process.

**Figure 2 pharmaceutics-15-01696-f002:**
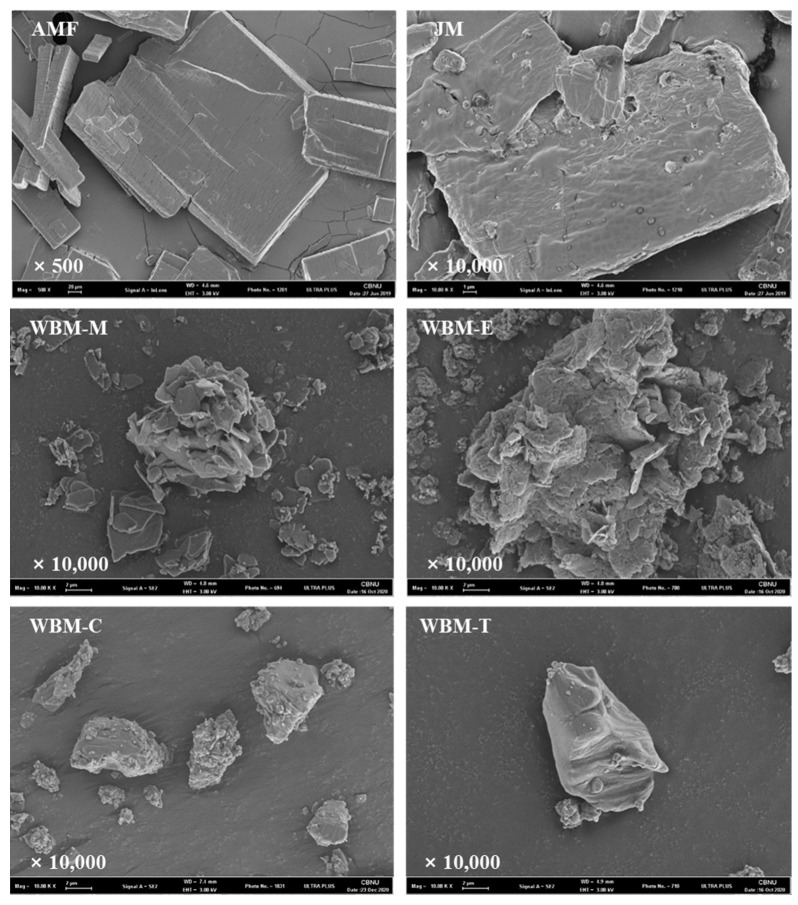
Scanning electron microscopy images of AMF, JM, WBM-M, WBM-E, WBM-C, and WBM-T. (Magnification: 500× and 10,000×).

**Figure 3 pharmaceutics-15-01696-f003:**
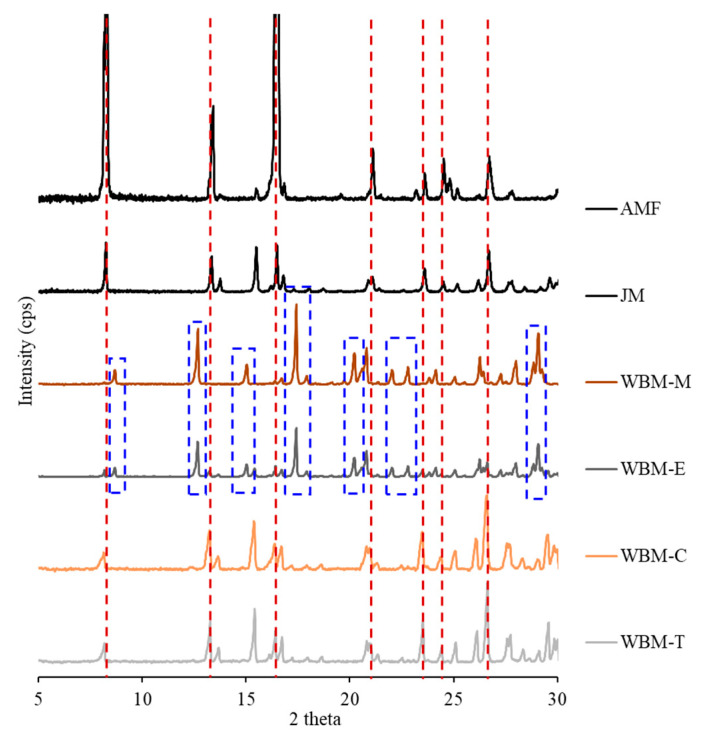
Powder X-ray diffraction (PXRD) of AMF, JM, WBM-M, WBM-E, WBM-C, and WBM-T; (Red line: AMF trihydrate main diffraction peaks, blue box: shifted peaks.).

**Figure 4 pharmaceutics-15-01696-f004:**
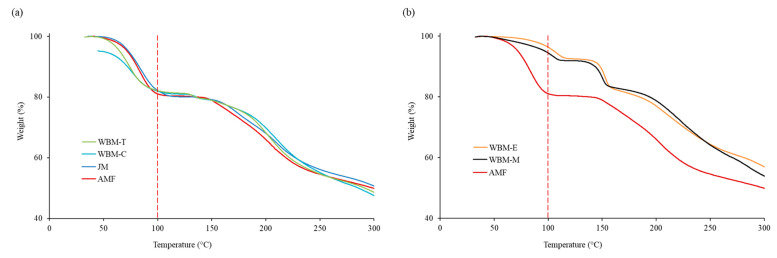
Thermal gravimetric analysis (TGA) of (**a**) AMF, JM, WBM-C, and WBM-T. (**b**) AMF, WBM-M, and WBM-E.

**Figure 5 pharmaceutics-15-01696-f005:**
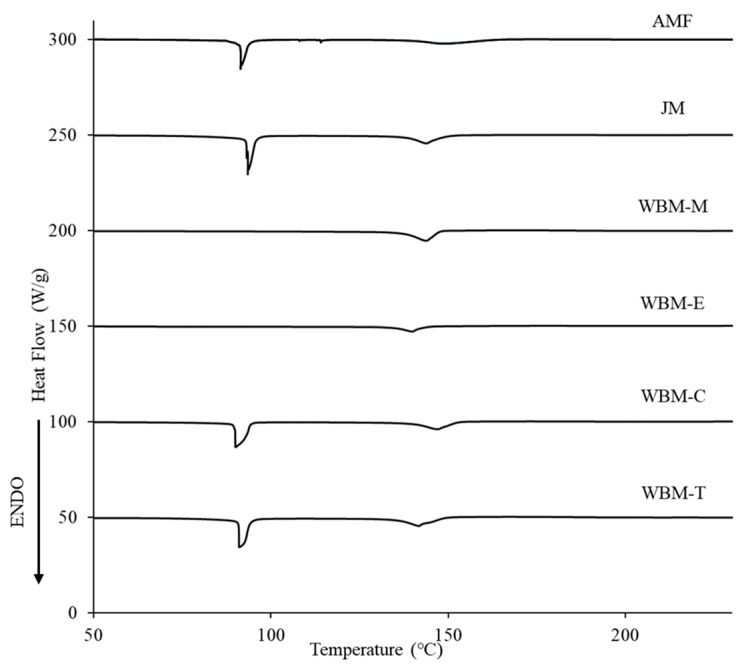
Differential scanning calorimetry (DSC) thermograms of AMF, JM, WBM-M, WBM-E, WBM-C, and WBM-T.

**Figure 6 pharmaceutics-15-01696-f006:**
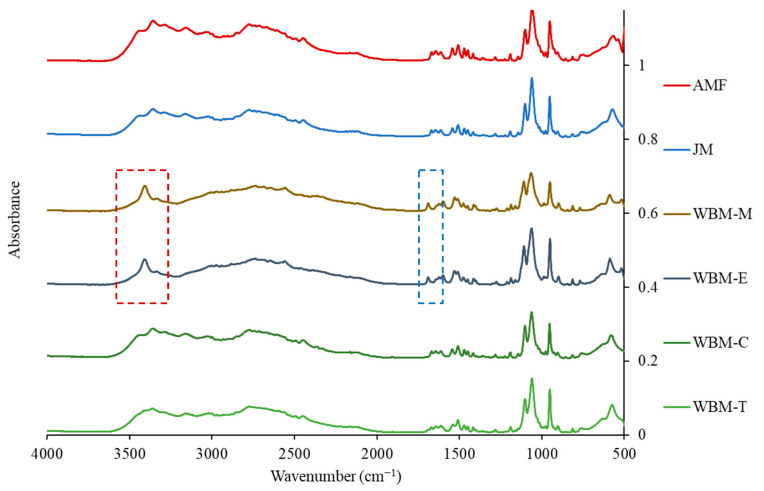
Fourier-transform infrared (FT-IR) spectra of AMF, JM, WBM-M, WBM-E, WBM-C, and WBM-T. (Red box: O-H stretching vibrations, blue box: C=O groups).

**Figure 7 pharmaceutics-15-01696-f007:**
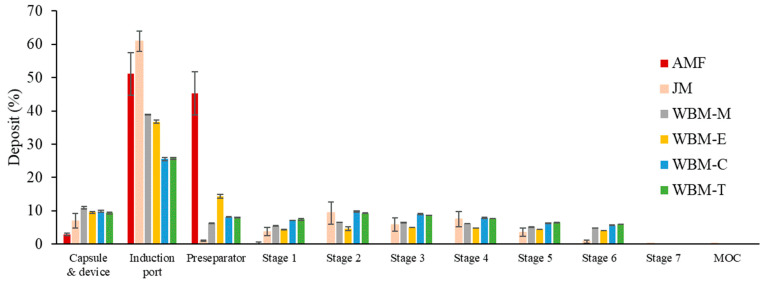
Percentage deposited in each stage of the Next-Generation Impactor for AMF, JM, WBM-M, WBM-E, WBM-C, and WBM-T (mean ± SD, n = 3).

**Table 1 pharmaceutics-15-01696-t001:** Formulation of ball-milled microparticles WBM-M, WBM-E, WBM-C, and WBM-T.

	WBM-M	WBM-E	WBM-C	WBM-T
Amifostine (g)	10	10	10	10
Bead (g)	50	50	50	50
Solvent (mL)	20	20	20	20
Solvent	Methanol	Ethanol	Chloroform	Toluene
Bead diameter (mm)	3	3	3	3
Rotation rate (RPM)	400	400	400	400
Rotation time (min)	15	15	15	15

## Data Availability

Data sharing not applicable.

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
