# Peer review of "Preparation and Evaluation of Inhalable Amifostine Microparticles Using Wet Ball Milling"

_pharmaceutics, 2023, doi:10.3390/pharmaceutics15061696_

Round 1

Reviewer 1 Report

It appears that publication in any form in “Pharmaceutics” journal would be premature at this time.

This paper developed the inhalable microparticles of amifostine (AMF) using different milling methods. The formulations were characterized and checked for its in vitro aerodynamic performance. However, current data is not enough to support the significance of method and its role in AMF application.

1) The ball milling method mentioned is not explained well.

2) Authors performed SEM to check the surface characterization. Why not atomic force microscopy (AFM)? AFM has been commonly used to distinguish polymorphs of many organic molecules and APIs. SEM images provided by the authors do not justify the claims made in the results. E.g. WBM-C surface is also looking rough.

3) Section 3.1.2. is mentioning each value from table 1. What is the use of table then! Also, authors do not explain what is Dv 10, 50 and 90 in table.

4) Section 3.1.5. mentioning “The melting point of AMF was 149℃ and of trihydrate 236 85℃–90℃”. However, there are no such peaks shown in figure 4. There is one peak below 100 ℃, but not annotated the exact value (85-90oC ?). Also, there is a small peak between 130-150 oC in all thermograms. No explanation for that.

5) No annotation in figure 5. No discussion on how FTIR is helpful to prove polymorphism other than proof of hydration.

6) How aerodynamic performance test supporting the objective “effective at low dose” (line 13)?

7) No data on how DPI formulation of AMF is more effective than IV Ethyol.

8) The discussion section is so lame. There is no proper explanation on how solvent may affect the polymorphism of APIs with references.

9) Line 293 mentioned WBM-C and T showed the same molecular weight. Where are molecular weights?

10) The language used in whole manuscript is vague.

The language used in whole manuscript is vague

Author Response

Thank you for your thorough review and salient observations of this manuscript and for the comments and suggestions, which help to improve the quality of this manuscript. Our response follows.

Reviewer 2 Report

This study developed inhalable microparticles of amifostine (AMF) and compared the physicochemical properties and inhalation efficiency of the AMF microparticles prepared by different methods (Jet milling & wet ball mill) and different solvents (methanol, ethanol, chloroform and toluene). The finding is of interest and significance. However, following minor changes need to be done.

1.      Different solvents including methanol, ethanol, chloroform and toluene were used for preparing AFM microparticles, was there any residual organic solvent present?

2.      No filtration was performed to remove the solvent before oven drying?

3.      Conclusions need to be more revealing with quantitative results data.

4.      There were some confusing expressions in the context. For example, â‘  Page 1 line 37&38 “The short half-life results in high concentrations of the drug”, â‘¡ Page 2 line 55&56 “Various types of mills can be used to reduce particle size.” However, followed description was different preparation types (both top-down and bottom-up), rather than various mills; â‘¢ “To prepare AMF particles, wet ball milling is demonstrated to have a better inhalation efficiency than jet milling.” It should be “AMF particles prepared by wet ball milling is demonstrated to have a better inhalation efficiency than that prepared by jet milling.”

Author Response

(The authors gave the same response as above.)

Reviewer 3 Report

The proposed study demonstrated the preparation and characterization of of inhalable amifostine (AMF) microparticles by ball milling method. What is the novelty here?

The authors did not demonstrate the development, validation and optimization of the ball milling method, which is the most important research of this manuscript. 

Abstract should be rewritten with additional data on the ball milling methods for manufacturing the AMF microparticles.

Introduction section needs to be rearranged focusing on the purpose of this study. The milling methods should be included in the objective of this study. 

It would be good to add a a schematic diagram of the ball milling method in the methodology section.

Line 16: what do you mean by JM?

Lines 17-17,  are the % values FPFs? Please rewrite the sentence.

Figure 4: Melting is an "endothermic" phenomena and  the direction of the peaks look opposite in the figure. An arrow sign (y axis) is required to ensure the direction of the  endothermic peaks.

Please present the % of amorphous and crystalline particles generated by jet and ball milling method.  

sections 2.2 and 2.3 Preparation of inhalable AMF microparticle using jet milling and ball milling methods, respectively. An appropriate clarifications are required to use these two methods.

Discussion section: A lot of data are generated; however, not appropriately explained in the discussion section.  Authors must appropriately explain all experimental data here linking the figures/table?. 

Conclusion should be rewritten 

Moderate editing of English is required

Author Response

(The authors gave the same response as above.)

Round 2

Reviewer 1 Report

The authors have addressed all concerns raised. I recommend this revised version to be considered for publication